# Systematic Review of Antimicrobial Combination Options for Pandrug-Resistant *Acinetobacter baumannii*

**DOI:** 10.3390/antibiotics10111344

**Published:** 2021-11-03

**Authors:** Stamatis Karakonstantis, Petros Ioannou, George Samonis, Diamantis P. Kofteridis

**Affiliations:** Department of Internal Medicine & Infectious Diseases, University Hospital of Heraklion, 71110 Heraklion, Crete, Greece; petros_io@hotmail.com (P.I.); samonis@med.uoc.gr (G.S.); kofterid@med.uoc.gr (D.P.K.)

**Keywords:** *Acinetobacter*, pandrug-resistant, antimicrobial combinations, synergy

## Abstract

Antimicrobial combinations are at the moment the only potential treatment option for pandrug-resistant *A. baumannii*. A systematic review was conducted in PubMed and Scopus for studies reporting the activity of antimicrobial combinations against *A. baumannii* resistant to all components of the combination. The clinical relevance of synergistic combinations was assessed based on concentrations achieving synergy and PK/PD models. Eighty-four studies were retrieved including 818 eligible isolates. A variety of combinations (*n* = 141 double, *n* = 9 triple) were tested, with a variety of methods. Polymyxin-based combinations were the most studied, either as double or triple combinations with cell-wall acting agents (including sulbactam, carbapenems, glycopeptides), rifamycins and fosfomycin. Non-polymyxin combinations were predominantly based on rifampicin, fosfomycin, sulbactam and avibactam. Several combinations were synergistic at clinically relevant concentrations, while triple combinations appeared more active than the double ones. However, no combination was consistently synergistic against all strains tested. Notably, several studies reported synergy but at concentrations unlikely to be clinically relevant, or the concentration that synergy was observed was unclear. Selecting the most appropriate combinations is likely strain-specific and should be guided by in vitro synergy evaluation. Furthermore, there is an urgent need for clinical studies on the efficacy and safety of such combinations.

## 1. Introduction

Pandrug-resistant (PDR) Gram-negative bacteria, resistant to all currently available antibiotics, including carbapenems, aminoglycosides, polymyxins and tigecycline, have been increasingly reported worldwide [1]. Especially problematic is the management of infections by PDR *A. baumannii* (PDRAB), since there are no monotherapy treatment options and associated mortality is very high [2]. Cefiderocol, where available, is a last resort option [3]. However, resistance to cefiderocol is already being reported and is likely to increase, considering the high prevalence of heteroresistance to this agent [4], as has occurred with polymyxins [5]. Therefore, pending approval of new antimicrobials, synergistic combinations are at the moment the only potential treatment option for PDRAB [6].

Combination antimicrobial therapy compared to monotherapy has not so far been proven in most studies to lead to better clinical outcomes of *A. baumannii* infections [7,8,9,10,11]. However, the available studies are predominantly based on combinations including at least one active antimicrobial and a potential benefit in PDRAB infections, with no monotherapy treatment options, should not be excluded [6,12,13]. Similar to clinical studies, prior systematic reviews that have assessed the in vitro synergy of various combinations (based on polymyxins [14,15,16], rifampin [14,16], meropenem [16,17] or tigecycline [16,18]) against *A. baumannii*, were predominantly based on studies testing combinations including at least one active antimicrobial. However, synergy testing may be most useful to identify combinations for salvage therapy of infections by bacteria resistant to all monotherapy treatment options [19].

Therefore, the purpose of this systematic review is to identify synergistic combinations that may be used for treatment of infections caused by PDRAB, i.e., combinations based on antimicrobials to which *A. baumannii* is resistant. Furthermore, it was evaluated whether the identified combinations were synergistic at concentrations achievable in vivo, a major consideration when assessing the in vivo relevance of in vitro synergy [20], especially when referring to PDRAB. These data aim to aid microbiology laboratories and infectious disease clinicians to prioritize the potential combination options for evaluation for synergy against the local PDRAB strains.

## 2. Methods

### 2.1. Search Strategy

The following search was conducted in PubMed from inception to 20 April 2021: (*Acinetobacter* [ti] OR baumannii [ti] OR “*Acinetobacter*” [Mesh] OR “*Acinetobacter baumannii*” [Mesh]) AND (synerg* [ti] OR combin* [ti] OR “Drug Combinations” [Mesh] OR “Drug Synergism” [Mesh] OR “Drug Therapy, Combination” [Mesh]). The same search, without the MESH terms, was also conducted in Scopus.

### 2.2. Eligibility Criteria

Any study (including in vitro, animal models, and clinical studies) evaluating the activity of antimicrobial combinations against clinical *A. baumannii* isolates was eligible, provided that the *A. baumannii* isolates tested were resistant to all components of the antimicrobial combinations assessed. The following exclusion criteria were applied: (1) studies including only noneligible isolates (see below definition for eligibility), (2) studies including both eligible and noneligible isolates, but not possible to extract data for eligible isolates, (3) combinations of antimicrobials with adjuvant, nonantibiotic agents, or with investigational agents (not currently in use for the treatment of infections). (4) Clinical studies without any information on synergy. (5) Studies written in languages other than English (little impact [21,22], often at higher risk of bias [23], and data extraction can be inaccurate [23]). Deduplication and screening for eligibility of the retrieved articles was conducted by the first author using the Rayyan online platform [24].

### 2.3. Data Extraction

The following data were extracted from each eligible article: country where the study was conducted, number of participating hospitals, methods of synergy testing (readers are referred to relevant references for a more detailed overview of the different methods [19,20,25,26,27]), list of antimicrobials tested for synergy, number of eligible strains (as defined below), number of eligible strains against which each combination demonstrated synergy and antimicrobial concentrations achieving synergy. Data were extracted by the first author in duplicate.

### 2.4. Definition of Eligible Strains

*A. baumannii* isolates were eligible for this review if resistant to all components of the antimicrobial combinations tested. The following breakpoints were used to define resistance based on CLSI [28] or EUCAST [29] clinical breakpoints (whichever was higher): amikacin > 32 mg/L, ampicillin-sulbactam > 16/8 mg/L, cefepime > 16 mg/L, cefiderocol > 8 mg/L, ceftazidime > 16 mg/L, ciprofloxacin > 2 mg/L, colistin > 2 mg/L, gentamicin > 8 mg/L, imipenem > 4 mg/L, levofloxacin > 4 mg/L, meropenem > 8 mg/L, minocycline > 8 mg/L, piperacillin > 64 mg/L, piperacillin/tazobactam > 64/4 mg/L, polymyxin B > 2 mg/L, tobramycin > 8 mg/L, trimethoprim-sulfamethoxazole > 2/38 mg/L. For antibiotics without established breakpoints by either EUCAST or CLSI the following cut-offs were applied: azithromycin > 4 mg/L (based on CLSI breakpoints for *Staphylococci* [12,28]), aztreonam >16 mg/L (based on breakpoints for *P. aeruginosa* [28,29]), cefoperazone/sulbactam > 32/16 mg/L [30], ceftazidime/avibactam > 8/4 mg/dl (based on breakpoints for *P. aeruginosa* [28,29]), chloramphenicol > 16 mg/L (based on breakpoints for Enterobacterales [28]), fosfomycin > 32 mg/L (based on EUCAST breakpoints for Enterobacterales and *Staphylococcus* spp [29]), fusidic acid > 1 mg/L (based on EUCAST breakpoints for *Staphylococcus* spp [29]), moxifloxacin > 0.25 mg/L (based on EUCAST breakpoints for Enterobacterales [29]), plazomicin > 4 mg/L (FDA interpretive criteria for Enterobacteriaceae [31]), rifampicin > 2 mg/L (based on CLSI breakpoints for *Staphylococci* [28], although much lower cut-offs have been proposed for *A. baumannii* [32]), tigecycline > 2 mg/L [33], trimethoprim > 8 mg/L (based on CLSI breakpoints for Enterobacterales [28]), vancomycin > 20 mg/L (based on clinically achievable concentrations [34,35,36], noting that the CLSI breakpoints for coagulase-negative *Staphylococci* is > 16 mg/L [28]).

### 2.5. Evaluation of In Vivo Feasibility of the Identified Combinations

In vivo feasibility of each synergistic combination was assessed based on the following: (1) synergy present in vitro at concentrations equal to or lower than established breakpoints of resistance (as defined above) for all antimicrobials used in the combination, or (2) synergy demonstrated in dynamic drug concentration-time experiments (such as the hollow-fiber infection model, or animal infection models) simulating the pharmacokinetics of human treatment regimens, or (3) clinically-achievable synergy based on pharmacokinetic/pharmacodynamic (PK/PD) modelling and Monte Carlo simulations [37].

### 2.6. Data Synthesis and Analysis

A qualitative synthesis of the data was conducted. Meta-analysis of the data was not pursued (a post hoc decision), based on the following findings of the review; methodological heterogeneity in synergy testing methods and interpretation, small number of studies and eligible isolates per combination, clonal relatedness of *A. baumannii* isolates from single-center studies, potential differences between different *A. baumannii* strains (i.e., synergy against *A. baumannii* strains isolated from one institution does not necessarily predict synergy against different strains, with different mechanisms and level of resistance), potential for publication bias (studies with negative results are less likely to be published), selective performance of more cumbersome synergy testing methods (such as time-kill assay or animal models) only against strains for which synergy had been demonstrated by other methods (such as checkerboard), questionable clinical relevance of synergy in many studies (synergy present only at high antimicrobial concentrations, likely not relevant for in vivo use, or at unclear concentrations).

## 3. Results

### 3.1. Summary and Characteristics of Reviewed Studies

A flow chart of the review is depicted in Figure 1. Eighty-four relevant publications [12,35,36,37,38,39,40,41,42,43,44,45,46,47,48,49,50,51,52,53,54,55,56,57,58,59,60,61,62,63,64,65,66,67,68,69,70,71,72,73,74,75,76,77,78,79,80,81,82,83,84,85,86,87,88,89,90,91,92,93,94,95,96,97,98,99,100,101,102,103,104,105,106,107,108,109,110,111,112,113,114,115,116,117] were retrieved including 818 eligible *A. baumannii* isolates. The characteristics of the reviewed studies are summarized in the Appendix A. Most (73%) studies were published in the last 10 years, while about a third (35%) were published in the last 5 years (Appendix B, Table A1). Most studies were conducted in the European region (33%), America (29%) and the Western-Pacific region (24%) (Appendix B, Table A2). The number of eligible isolates per study was small in most studies, with most (79%) of them including ≤ 10 isolates (Appendix A. Finally, most studies were single center (65%) and of the multicenter studies most (58%) were conducted in only two to five centers (Appendix A, an important consideration as this reflects the clonal diversity of the *A. baumannii* isolates available for each study.

### 3.2. Overview of Methods for Assessment of Antimicrobial Combinations

A variety of methods were used for in vitro evaluation of antimicrobial combinations; disk diffusion methods (*n* = 4 studies, *n* = 18 eligible isolates), gradient strip methods (*n* = 11 studies, *n* = 229 eligible isolates), MIC determination by agar dilution (*n* = 2 studies, *n* = 42 eligible isolates), checkerboard assay (*n* = 44 studies, *n* = 599 eligible isolates), the multiple-combination bactericidal test (*n* = 1 study, *n* = 9 eligible isolates), time-kill assay (*n* = 51 studies, *n* = 259 eligible isolates), dynamic in vitro PK/PD models with antimicrobial concentrations simulating human treatment regimens (*n* = 6 studies, *n* = 10 isolates), and semi-mechanistic PK/PD modelling based on TKA data (*n* = 5 studies [37,54,102,107,118]). Finally, a few in vivo animal models (*n* = 11 studies, *n* = 18 isolates) eligible for review have been published [35,38,55,64,70,90,94,98,102,105,113]. No eligible clinical studies were retrieved.

### 3.3. Overview of Antimicrobial Combinations That have been Evaluated

Numerous different combinations (*n* = 141 double and *n* = 9 triple combinations) were evaluated predominantly based on polymyxins, rifamycins (predominantly rifampicin and recently rifabutin), sulbactam, fosfomycin and carbapenems. However, there were few available studies for most combinations with only 10 combinations having >3 studies available. Summarizing Tables of the number of studies and number of eligible isolates for each combination, as well as methods used to evaluate each combination are available in the Appendix A.

### 3.4. Overview of Polymyxin-Based Combinations

Polymyxin-based combinations were the most studied, with several studies demonstrating synergy against eligible *A. baumannii* isolates by combinations of polymyxins (either colistin or polymyxin-B) with cell-wall acting agents including: sulbactam (either alone or as ampicillin-sulbactam), beta-lactams (predominantly carbapenems, but also third generation cephalosporins, aztreonam, and ceftazidime/avibactam), glycopeptides (predominantly vancomycin, but also teicoplanin), and daptomycin. Furthermore, several studies have reported synergy between colistin and rifamycins against eligible strains (predominantly rifampicin and recently rifabutin). Isolated reports have also demonstrated synergy with trimethoprim/sulfamethoxazole, chloramphenicol, and fusidic acid.

The following triple polymyxin-based combinations have also been shown to be synergistic against selected eligible strains: polymyxin-B/meropenem/sulbactam [51,69], polymyxin-B/meropenem/ampicillin/sulbactam [61,62], colistin/doripenem/sulbactam [82], polymyxin-B/meropenem/fosfomycin [51,69] and polymyxin-B/doripenem/vancomycin [35]. Triple polymyxin-based combinations appear to be more active than double combinations and more likely to prevent regrowth during treatment [51,61,69,82], likely by preventing emergence of resistant subpopulations [61].

A variety of the above combinations (colistin/sulbactam, polymyxin-b/sulbactam, colistin/imipenem, colistin/meropenem, polymyxin-B/meropenem, colistin/doripenem, colistin/tigecycline, colistin/rifampicin, polymyxin-B/rifampicin, colistin/vancomycin, polymyxin-B/vancomycin, colistin/daptomycin, colistin/trimethoprim/sulfamethoxazole, colistin/chloramphenicol, colistin/fusidic acid, colistin/levofloxacin, polymyxin-B/fosfomycin/meropenem, polymyxin-B/sulbactam/meropenem, polymyxin-B/ampicillin/sulbactam/meropenem, colistin/sulbactam/doripenem, colistin/vancomycin/doripenem) have been shown to be synergistic at concentrations equal to or less than established breakpoints by a variety of methods, or in dynamic drug concentration-time experiments including animal models (Appendix B; Table A3, Table A4 and Table A5, and Appendix A. Nevertheless, the number of studies and eligible isolates per combination was small and most combinations were active at clinically relevant concentrations only against selected of the tested eligible strains (Appendix B; Table A3, Table A4 and Table A5, and Appendix A.

### 3.5. Overview of Non-Polymyxin Based Combinations

Non-polymyxin-based combinations are predominantly based on combinations of the following antimicrobials (Appendix A: sulbactam (either as sulbactam alone or in the form of ampicillin/sulbactam or cefoperazone/sulbactam), fosfomycin, rifampicin and carbapenems. However, a variety of other antimicrobials have been tried in combination regimens including aminoglycosides, tetracyclines (doxycycline, tigecycline, minocycline and eravacycline), fluoroquinolones, cephalosporins, aztreonam, trimethoprim/sulfamethoxazole, linezolid, teicoplanin and azithromycin.

The best data for non-polymyxin-based combinations come from four studies by Mohd Sazly Lim S et al. [37,44,45,118]. Fosfomycin/sulbactam (FOF/SUL), fosfomycin/meropenem (FOF/MEM), sulbactam/meropenem (SUL/MEM), fosfomycin/rifampin (FOF/RIF) and meropenem/rifampin (MEM/RIF) were evaluated for synergy against 50 eligible *A. baumannii* isolates characterized by high genetic diversity. The combinations were first evaluated by checkerboard assay [44]. Based on an FICI ≤ 0.5 the combinations were synergistic against 74% (FOF/SUL), 28% (FOF/MEM), 56% (SUL/MEM), 24% (FOF/RIF) and 20% (RIF/MEM) of eligible strains. Synergy was mostly detected at concentrations above established breakpoints of resistance. However, considering higher proposed breakpoints based on PK/PD models (32 mg/L for SUL [119,120] and 128 mg/L for FOF [45,121]) the combination FOF/SUL was active against 18 of 28 (64%) eligible isolates [37], the combination FOF/MEM was active against 9 of 33 (27%) eligible isolates [45], and the combination SUL/MEM was active against 9 of 46 (20%) eligible isolates [118]. FOF/SUL and SUL/MEM were further evaluated in TKA against selected isolates [37,44,118], but synergy was only reported at concentrations (128/128 mg/L for SUL/FOF and 64/32–128/64 for SUL/MEM) higher than established breakpoints.

Finally, Mohd Sazly Lim S et al. evaluated two of the above combinations with semi-mechanistic PK/PD modelling; FOF/SUL (simulated regimen: 8 g of fosfomycin given every 8 h as a 1 h infusion and 4 g of sulbactam given every 8 h as a 4 h infusion) [37] and SUL/MEM (simulated regimen: 2 gr of meropenem given every 8 h as a 3 h infusion, and 4 g of sulbactam given every 8 h as a 4 h infusion [118]). A high probability of target attainment was shown for FOF/SUL against the selected isolate (FOF MIC 2048, SUL MIC 128, combination MIC in checkerboard 32/16 mg/L); 81.6%, 76.4%, and 71.6% for stasis, 1-log_10_ kill and 2-log_10_ kill, respectively (compared to 23.3%, 19.8% and 15.5% for fosfomycin monotherapy, and 53.5%, 46.5%, and 32.5% for sulbactam monotherapy) [37]. In contrast, the probability of target attainment was at best moderate for SUL/MEM against the selected isolates (MEM MIC 128 mg/L, SUL MIC 256 mg/L, combination MICs 8/64 and 8/32 mg/L); 41%, 38% and 34% for stasis, 1-log_10_ kill and 2-log_10_ kill, respectively (compared to no killing with either of the monotherapies) [118].

Avibactam/sulbactam is another recently proposed promising combination. Rodriguez CH et al. [47] showed that avibactam at a fixed concentration of 4 mg/L reduced the MIC of sulbactam to ≤4 mg/L in all 35 non-metallo-β-lactamase (MBL)-producing sulbactam-resistant *A. baumannii* isolates in one study. The activity of sulbactam/avibactam (and to a lesser extent of sulbactam/relebactam) was also confirmed in a subsequent study [122]. The rationale of the combination is that avibactam may inhibit the β-lactamases that affect activity of sulbactam [47]. However, the combination is less effective against metallo-β-lactamase-producing isolates [47,122].

In contrast to non-MBL Enterobacterales [6], double carbapenem combinations are less likely to be clinically relevant for *A. baumannii* strains. Specifically, the combination meropenem/imipenem was synergistic against 6 of 21 eligible isolates according to checkerboard assay in one study, but synergy was only observed at concentration above established breakpoints of resistance (synergy was present at the following meropenem/imipenem concentrations: 16/4, 16/8, 32/16 and 32/32, 16/8 mg/L) and all isolates had relatively low MICs (mostly 32–64 mg/L) [46]. The combination imipenem/meropenem has also been shown to be effective in a murine intraperitoneal infection model (using two *A. baumannii* strains with meropenem-imipenem MICs 16–16 and 32–32 mg/L, respectively), but mortality and bacterial clearance were similar comparing meropenem monotherapy to combination therapy [38]. Additionally, the combination imipenem/ertapenem was not found to be synergistic in another study [73].

### 3.6. Evaluation of Clinical Relevance of Reported Synergy

Detailed data regarding the proportion of observed synergy for each combination (per study and method) and assessment of the clinical relevance are available in the Appendix A. In most cases, synergy was only reported at antimicrobial concentrations above the established breakpoints of resistance or the concentration at which synergy was observed was not reported. Specifically, of *n* = 539 cases of reported synergy in checkerboard assay, synergy was observed at concentrations ≤breakpoints in only 112 (21%) cases, synergy was reported at concentration >breakpoints in 194 (36%) cases, while in 233 (43%) cases the concentration at which synergy was present was unclear. Similarly, of *n* = 185 cases of reported synergy in TKA, synergy was observed at concentrations ≤breakpoints in only 65 (35%) cases, synergy was reported at concentration >breakpoints in 88 (48%) cases, while in 32 (17%) cases the concentration at which synergy was present was unclear.

Additionally, the clinical relevance of improved outcomes (survival, reduction of bacterial loads, sterilization of cultures) in animal models is unclear, despite simulation of human treatment regimens, considering the unexpectedly high efficacy of monotherapies in many cases [38,90,94,98,105,113], and potentially nonrelevant for humans mechanisms of action of antimicrobials [35]. Finally, dynamic in vitro PK/PD models [61,62,73,87,88,107] and semi-mechanistic PK/PD models were available for only a few combinations and selected isolates [37,54,102,107,118] but provided useful information about the killing activity of antimicrobial combinations at clinically relevant concentrations.

A summary of combinations that have been found synergistic at concentrations ≤established breakpoints of resistance are available in Table A3 of Appendix B. Studies using dynamic in vitro PK/PD models or animal models are summarized in Table A4 and Table A5 of Appendix B.

### 3.7. Clinical Studies

Although several studies have assessed antimicrobial combination in *A. baumannii* infections (e.g., [7,8,9,10,11,123,124]) none was eligible for this review for the following reasons: (a) combinations were assessed in patients with noneligible isolates (i.e., isolates susceptible to at least one component of the combination) or the extraction of data for eligible isolates was not possible, and/or (b) lack of in vitro evaluation for the presence of synergy. The latter is important because, as demonstrated in this review, in vitro synergy observed against selected *A. baumannii* strains with a specific combination cannot be generalized to other *A. baumannii* strains. Furthermore, the very few available studies including patients with infections by PDRAB [1,6,124,125] have major limitations, including small study populations, retrospective designs, lack of a control group or direct comparison of different treatment regimens, and lack of correlation of in vitro susceptibility testing of the combinations with outcomes.

Notable among the available studies is a secondary analysis of the AIDA study (a randomized controlled trial comparing colistin monotherapy to colistin-meropenem combination in patients with carbapenem-resistant Gram-negative infections [9]) comparing monotherapy to combination therapy against colistin- and carbapenem-resistant *A. baumannii* infections [10]. Based on this study, the colistin-meropenem combination was paradoxically associated with higher mortality compared to monotherapy [10]. However, being an exploratory subgroup analysis, the study has several limitations and data on the presence (or absence) of synergy were not reported for the subgroup of patients with colistin- and carbapenem-resistant *A. baumannii* infections. Nevertheless, the study raises the hypothesis that blindly (in the absence of clinical data) using antimicrobial combinations could unexpectedly result in worse outcomes.

In contrast, favorable results have been reported in a few small series (with all the above-mentioned limitations) with selected combinations. For example, triple combination therapy with high-dose ampicillin/sulbactam, high-dose tigecycline and colistin in patients with ventilator-associated pneumonia by PDRAB resulted in clinical cure in 9 of 10 patients [125]. Similarly, in another series, all seven patients with ventilator-associated pneumonia or bacteremia by colistin-resistant *A. baumannii* were successfully treated with a triple combination including colistin, doripenem and ampicillin/sulbactam (although with one exception, all isolates had ampicillin/sulbactam MICs ≤ 16/8 mg/L, i.e., were not eligible for this review) [126]. Furthermore, the combination of colistin with rifampicin has been used successfully to treat post-neurosurgical meningitis after emergence of colistin resistance during treatment with colistin monotherapy [127,128]. However, eligibility of the included isolates in the latter studies could not be assessed due to lack of reporting of rifampicin MICs [127,128].

Therefore, clinical studies assessing antimicrobial combinations in infections by PDRAB are urgently needed. The selection of antimicrobial combinations for further clinical study should ideally be guided by in vitro susceptibility testing of the combinations against local *A. baumannii* strains, taking into account whether synergy is achievable at clinically relevant concentrations.

## 4. Discussion

### 4.1. Summary of Main Findings

The emergence of XDR/PDR *A. baumannii* [1], which is associated with high mortality [2] and limited treatment options [6], has resulted in an increasing number of publications evaluating the role of antimicrobial combination therapy. A vast number of potential combinations has been reported, although most combinations have been evaluated only against a limited number of eligible *A. baumannii* isolates. The most studied combinations are polymyxin-based combinations with cell-wall acting agents (including sulbactam, carbapenems and vancomycin), rifampicin and fosfomycin. Nevertheless, a variety of combinations have been reported to be synergistic at clinically achievable concentrations, at least against selected *A. baumannii* isolates. However, in most cases synergy was reported either at too high concentrations or at unclear concentrations.

### 4.2. Polymyxin-Based Combinations

Polymyxin-based combinations were originally proposed to prevent treatment failure due to the emergence of polymyxin-resistant *A. baumannii* during therapy [129], but may actually be most useful for PDRAB [5,125,127,128]. A proposed mechanism to explain the synergy between polymyxins and other antimicrobials is that polymyxins, even at subinhibitory concentrations, may increase the permeability of *A baumannii*’s cell wall to other antimicrobials, including antimicrobials that would otherwise be ineffective against Gram-negative pathogens (such as glycopeptides and lipopeptides) [12,34,56,88].

Polymyxins may be combined, either as double or as triple combinations, with a variety of antimicrobials, including carbapenems, sulbactam, fosfomycin, rifampicin, rifabutin (which has recently been shown to be much more potent than rifampicin [130] and may retain activity even against PDRAB [131]) and vancomycin. Synergy with many of these combinations was achievable at concentrations ≤established breakpoints of resistance and demonstratable in animal models and/or dynamic in vitro PK/PD studies simulating human treatment regimens.

However, synergy is not universal and not applicable to every *A. baumannii* strain. Clinically relevant synergy may be less likely for strains with very high MICs. For example, clinically-relevant synergy between polymyxins and carbapenems appears to be less likely for isolates with high carbapenem MIC (doripenem >64 mg/L [82], meropenem ≥64 mg/L [132]). Triple combinations may be more effective than double combinations, by lowering MICs of individual agents to even lower levels and preventing emergence of resistance during treatment [51,61,69,82].

### 4.3. Non-Polymyxin Combinations

A variety of non-polymyxin combinations have been reported, predominantly involving the following antimicrobials: carbapenems, fosfomycin, sulbactam and rifamycins. The combination fosfomycin/sulbactam and to a lesser extent meropenem/sulbactam are especially promising and most studied [37,44,118], but a variety of other combinations have been found synergistic against selected eligible *A. baumannii* isolates. Such combinations may be even more active as triple combinations with polymyxins [51,61,69,82]. Furthermore, among non-polymyxin combinations, the recently proposed avibactam/sulbactam combination (aiming to restore susceptibility to sulbactam by inhibition of non-MBL β-lactamases with avibactam) is particularly promising and warrants further study [47,122].

Tigecycline-based combinations are often used in clinical practice against PDRAB [124,133], probably because of MICs closer to the cut-off for susceptibility [12]. However, based on the limited available data, tigecycline-based (or other tetracyclines, including eravacycline and minocycline) combinations are seldomly synergistic against resistant *A. baumannii* strains at clinically achievable concentrations [12,53,63,71,77,89,96,103,104,117]. However, the lack of in vitro synergy does not preclude a role for tigecycline in the treatment of XDR/PDR *A. baumannii*, especially with higher dose regimens that are predicted to achieve PK/PD targets for isolates with MICs up to 4–8 mg/L [134].

### 4.4. Limitations of the Review and of the Available Evidence

Despite the abundance of in vitro studies evaluating a variety of antimicrobial combinations against XDR/PDR *A. baumannii*, in vivo data, PK/PD models and clinical data are still limited. Furthermore, there is no acceptable gold standard method (one that best predicts in vivo efficacy) for the in vitro evaluation of synergy, mainly due to the lack of studies correlating in vitro synergy to clinical outcomes [19], and the results of different methods are often conflicting [25,68].

Moreover, as demonstrated in this review, studies often fail to assess the clinical relevance of reported synergy, as evidenced by the evaluation for synergy at antimicrobial concentration unlikely to be clinically relevant or lack of reporting of concentrations at which synergy is present. For example, an FIC index ≤ 0.5 in checkerboard assay does not necessarily prove clinically relevant synergy if antimicrobials are synergistic at concentrations higher than those achievable in vivo at the site of the infection. Similarly, in time-kill assays antimicrobials should ideally be used in concentrations achievable at the site of infection [20], which is often not the case as demonstrated in this review.

However, although clinically-relevant synergy was defined as synergy achievable at concentrations ≤ breakpoints of resistance it should be acknowledged that potentially higher breakpoints have been estimated (based on PK/PD data and Monte Carlo simulations) for high-dose, prolonged-infusion regimens [6]. For example, a high probability of target attainment with such regimens has been reported up to the following maximum MICs: meropenem ≤128 mg/L [135], doripenem ≤8 mg/L [136], fosfomycin ≤128 mg/L [45,121], sulbactam ≤32 mg/L [119,120]. Furthermore, some studies have evaluated the feasibility of synergistic combinations based on maximum clinically achievable concentrations [44,59] but we believe this approach could result in overestimating the in vivo relevance of synergistic combinations. Finally, the clinical relevance of synergy in animal models, even when using dosing regimens simulating human pharmacokinetics, is unclear considering that in some studies high efficacy was seen even for monotherapies against resistant strains [38,113], while in some cases antimicrobials may have additional functions in animal models not relevant to humans [35].

Finally, another major limitation of this review is the limited clonal diversity of eligible *A. baumannii* isolates for most combinations evaluated, considering that most studies were single-center and that for most combinations only few eligible isolates were assessed. This, combined with the inconsistent activity of antimicrobial combinations highlight the need to confirm in vitro synergy against local *A. baumannii* strains before using any of these combinations in clinical practice.

### 4.5. Strengths of the Review

Despite the above limitations, this is an exhaustive review of antimicrobial combination options against PDRAB, aiming to aid clinicians, researchers and microbiology laboratories to prioritize the selection of the most promising combinations for further evaluation against PDRAB. Furthermore, a detailed assessment of the potential clinical relevance of each synergistic combination was conducted, based on the concentrations that synergy was observed and the availability of PK/PD or animal models.

## 5. Conclusions

Antimicrobial combinations may be the only treatment option against PDR *A. baumannii*. Numerous combinations have been evaluated and several appear to be active at clinically relevant concentrations, at least against selected eligible *A. baumannii* isolates. However, studies often do not report the concentrations at which synergy is observed or use antimicrobials at concentrations unlikely to be clinically relevant. This is an important limitation of the available literature and an important consideration for future studies evaluating antimicrobial combinations against PDRAB. Furthermore, no combination was consistently synergistic against all isolates evaluated. Therefore, selecting the most appropriate combination is likely strain-specific and should be guided by in vitro synergy evaluation. Combinations demonstrating activity at clinically relevant concentrations and/or supported by PK/PD data and animal models should be further evaluated in appropriately designed clinical studies, which are currently lacking.

## Figures and Tables

**Figure 1 antibiotics-10-01344-f001:**
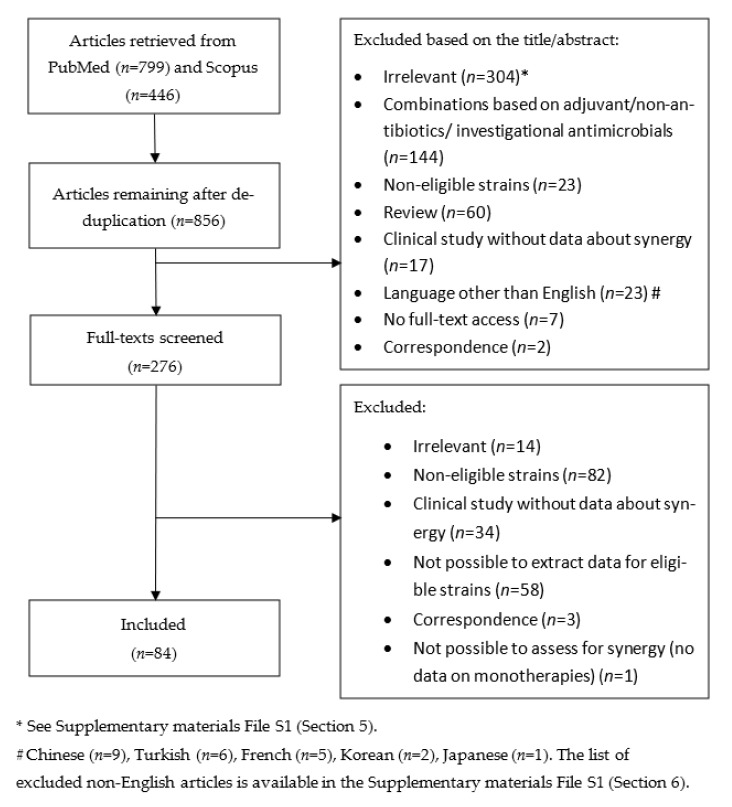
Flow chart of the review.

## Data Availability

The summary of characteristics and findings of each study included in this review is available in the Appendix A.

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
