# Peer review of "Systematic Review of Antimicrobial Combination Options for Pandrug-Resistant Acinetobacter baumannii"

_antibiotics, 2021, doi:10.3390/antibiotics10111344_

Round 1

Reviewer 1 Report

Authors performed an extensive study: "Systematic review of antimicrobial combination options for pandrug-resistant Acinetobacter baumannii"
Systematic review was conducted in PubMed and Scopus bases for studies reporting the activity of antimicrobial combinations against A. baumannii but which are resistant to all components of the combination. Total of 84 studies were retrieved including 818 eligible isolates.

Introduction is well written and proper references are inserted. Although, there is a statement which in my opinion should be changed. Authors state:
"Therefore, pending approval of new antimicrobials, synergistic combinations are the only potential treatment option for PDRAB"
It is not correct to say that this "synergistic combination are the only potential treatment". Authors should soften their claim with "may be" or "at the moment".

Page 2, 2.2. Eligibility Criteria
In Exclusion criteria: (5) Studies written in languages other than English.
What was the reasoning behind the Exclusion criteria? Why were these studies rejected?

Page 4, I am wandering about the different number of papers from PubMed and Scopus. Also,
there is a results that 304 papers were rejected as irrelevant based on the title/abstract.
This seems as a very large number in comparison with the total number of papers after duplication. What was the reason for that? Could the authors pinpoint which keyword was responsible for wrong paper identification that had to be excluded from the analysis? This would be valuable for potential readers.

Some of the results placed in the supplementary should be in the paper, e.g. Distribution of studies by year of publication and by country and WHO regions. These are interesting results and deserve to be in the paper.

Page 9, 4.4.
In Conclusion authors state:
"However, studies often do not report the concentrations at which synergy is observed or use antimicrobials at concentrations unlikely to be clinically relevant"

and they made a comment on page 9,
"Moreover, as demonstrated in this review, studies often fail to assess the clinical relevance of reported synergy, as evidenced by the evaluation for synergy at antimicrobial concentration unlikely to be clinically relevant or lack of reporting"

Authors clearly identified the main problems with the investigated previously published works. But at the same time this fact make their analysis scarce (without any attempt to diminish the quality and extent of the work). I think that it would be nice if the authors could clearly identify and present all the problems in the investigated papers. These problems should be clearly stated in the Conclusion (perhaps emphasized with bullets) and maybe even in the title of the paper. This could bring an additional weight to their results.

Author Response

We thank the reviewer for the insightful comments.

A point-by-point reply can be found below.

Point 1: Introduction is well written and proper references are inserted. Although, there is a statement which in my opinion should be changed. Authors state:

"Therefore, pending approval of new antimicrobials, synergistic combinations are the only potential treatment option for PDRAB"

It is not correct to say that this "synergistic combination are the only potential treatment". Authors should soften their claim with "may be" or "at the moment". 

Response 1: By definition there is no active monotherapy treatment option for pandrug-resistant A. baumannii. Nevertheless, we agree that in the future more options will be available. Therefore, we modified the text as suggested to: “Therefore, pending approval of new antimicrobials, synergistic combinations are at the moment the only potential treatment option for PDRAB”. The same modification was also made in the first sentence of the abstract: “Antimicrobial combinations are at the moment the only potential treatment option for pandrug-resistant A. baumannii

Point 2: Page 2, 2.2. Eligibility Criteria

In Exclusion criteria: (5) Studies written in languages other than English.

What was the reasoning behind the Exclusion criteria? Why were these studies rejected?

Response 2: This is already an extensive review of the literature and there is no reason to believe that including non-English literature would meaningfully change the results and conclusions of the manuscript. As stated in the Cochrane Handbook for Systematic reviews: “Several studies have found that in most cases there were no major differences between summary estimates of meta-analyses restricted to English-language studies compared with meta-analyses including studies in languages other than English”. Furthermore, data extraction from non-English articles can be inaccurate (https://doi.org/10.1186/2046-4053-2-97) and often non-English articles are of lower quality and at higher risk of bias (https://doi.org/10.1136/bmj.j2490).  Considering the above, trying to include non-English articles would substantially increase the workload of the review without meaningfully improving the manuscript. The text in Methods was modified as follows: “Studies written in languages other than English (little impact [21,22], often at higher risk of bias [23], and data extraction can be inaccurate [23])”. Furthermore, in the legend of Figure 1 readers are referred to the list of potentially relevant non-English articles excluded (Supplement, Section 6).

Point 3: Page 4, I am wandering about the different number of papers from PubMed and Scopus. Also, there is a results that 304 papers were rejected as irrelevant based on the title/abstract.This seems as a very large number in comparison with the total number of papers after duplication. What was the reason for that? Could the authors pinpoint which keyword was responsible for wrong paper identification that had to be excluded from the analysis? This would be valuable for potential readers.

Response 3: Having a different number of retrieved articles from different databases is not unexpected. The number of articles retrieved from Scopus was lower because of not including MESH terms in the search (in contrast to PubMed)(This is clearly stated in Methods: “The same search, without the MESH terms, was also conducted in Scopus”).

A large number of irrelevant articles is also not unexpected in my experience. The more sensitive a search strategy, the more the irrelevant articles. Considering the reviewer’s suggestion to explain which keywords were responsible for wrong paper identification a relevant paragraph was added in the Supplement (Section 5) and cited in the legend of Figure 1.

Point 4: Some of the results placed in the supplementary should be in the paper, e.g. Distribution of studies by year of publication and by country and WHO regions. These are interesting results and deserve to be in the paper.

Response 4: Although we agree that many of the results presented in the Supplement are important, the Supplement is too large. Including more Tables in the main manuscript would disrupt the flow of the manuscript. Considering the reviewer’s suggestion, the distribution of studies by year of publication and by country was added in the Appendix of the manuscript.           

Point 5: Page 9, 4.4.

In Conclusion authors state:

"However, studies often do not report the concentrations at which synergy is observed or use antimicrobials at concentrations unlikely to be clinically relevant"

and they made a comment on page 9,

"Moreover, as demonstrated in this review, studies often fail to assess the clinical relevance of reported synergy, as evidenced by the evaluation for synergy at antimicrobial concentration unlikely to be clinically relevant or lack of reporting"

Authors clearly identified the main problems with the investigated previously published works. But at the same time this fact make their analysis scarce (without any attempt to diminish the quality and extent of the work). I think that it would be nice if the authors could clearly identify and present all the problems in the investigated papers. These problems should be clearly stated in the Conclusion (perhaps emphasized with bullets) and maybe even in the title of the paper. This could bring an additional weight to their results.

Response 5: I am not sure I understand this point.

  • The reviewer suggests to “clearly identify and present all the problems in the investigated papers”. However, as the reviewer recognizes (“Authors clearly identified the main problems with the investigated previously published works”), the limitations of the review and of the available evidence are clearly discussed (4 paragraphs in section 4.4). Furthermore, the characteristics of each study, as well as data regarding the concentrations at which synergy was observed in each study and for each combination are presented in detail in the Supplement (Sections 2.1 and 4.4-4.7).
  • The reviewer suggests that “These problems should be clearly stated in the Conclusion”. However, the main limitations are already being acknowledged in the Conclusion:
    • “However, studies often do not report the concentrations at which synergy is observed or use antimicrobials at concentrations unlikely to be clinically relevant”
    • “should be further evaluated in appropriately designed clinical studies, which are currently lacking”.
  • Finally, the reviewer suggests stating the limitations “even in the title of the paper”. However, modifying the title to acknowledge the limitations of the literature is impractical. The main limitations are nevertheless clearly acknowledged in the Abstract (as much as possible considering word count restrictions):
    • “Notably, several studies reported synergy but at concentrations unlikely to be clinically relevant, or the concentration that synergy was observed was unclear”
    • “Furthermore, there is an urgent need for clinical studies on the efficacy and safety of such combinations.”

Reviewer 2 Report

the overall findings have been described nicely, Specifically, the limitations and strong outcomes have been summarized rationally. The authors could have included other terminologies in the search strategy, for example, antimicrobial peptides combined with conventional antibiotics.

Author Response

Point 1: the overall findings have been described nicely, Specifically, the limitations and strong outcomes have been summarized rationally. The authors could have included other terminologies in the search strategy, for example, antimicrobial peptides combined with conventional antibiotics.

Response 1: We thank the reviewer. As stated in Methods, studies evaluating “Combinations of antimicrobials with adjuvant, non-antibiotic agents, or with investigational agents (not currently in use for the treatment of infections)” were not eligible for this review. This includes antimicrobial peptides.

Reviewer 3 Report

 This study summarized the activity of antimicrobial combinations against A. baumannii resistant to all components of the combination. this paper analyzed a variety of combinations including polymyxin-based combination, non-polymyxin based combination and combinations with clinical relevant concentrations. This work can be accepted after illustrate the following questions:

  1. Address the importance for the treatment of PDRAB in the introduction and summary the present clinical status for the treatment of PDRAB to make it more clear the meaningful of your systematic review.
  2. English polish required 

minor mistakes:

1.line 37 ''may be'' to ''maybe''

2.line 116 " on the following;'' change to "on the following:"

3. line 275-276 " Notable....was not reported" reorganize the sentence, it is confusing.

Author Response

Point 1: This study summarized the activity of antimicrobial combinations against A. baumannii resistant to all components of the combination. this paper analyzed a variety of combinations including polymyxin-based combination, non-polymyxin based combination and combinations with clinical relevant concentrations. This work can be accepted after illustrate the following questions:

  1. Address the importance for the treatment of PDRAB in the introduction and summary the present clinical status for the treatment of PDRAB to make it more clear the meaningful of your systematic review.

Response 1: We thank the reviewer for the suggestion. However, I believe expanding further on PDRAB in the Introduction is beyond the scopes of this manuscript. Furthermore, by definition, there are no active monotherapy treatment options for pandrug-resistant A. baumannii. Interested readers are referred to relevant systematic reviews (e.g. reference 1 and reference 6). I believe the way the Introduction is currently written should be sufficient to convince the reader about the need for combination therapy for PDRAB considering:

(1) increasing prevalence of pandrug-resistant bacteria: “Pandrug-resistant (PDR) Gram-negative bacteria, resistant to all currently available antibiotics, including carbapenems, aminoglycosides, polymyxins and tigecycline, have been increasingly reported worldwide [1].”,

(2) high mortality attributable to infections by pandrug-resistant A. baumannii: “Especially problematic is the management of infections by PDR A. baumannii (PDRAB), since there are no monotherapy treatment options and associated mortality is very high [2]”,

(3) potential for emergence of resistance to last resort novel antimicrobials, including cefiderocol: “Cefiderocol, where available, is a last resort option [3]. However, resistance to cefiderocol is already being reported and is likely to increase considering the high prevalence of heteroresistance to this agent [4], as has occurred with polymyxins [5]

(4) lack of alternative treatment options: “Therefore, pending approval of new antimicrobials, synergistic combinations are at the moment the only potential treatment option for PDRAB [6].

Point 2: English polish required

Response 2: The manuscript was re-checked to correct any mistakes

Point 3: line 37 ''may be'' to ''maybe''

Response 3: “may be” was changed to “is” to avoid confusion

Point 4: line 116 " on the following;'' change to "on the following:"

Response 4: Done       

Point 5: line 275-276 " Notable....was not reported" reorganize the sentence, it is confusing.

Response 5: The sentence was modified as follows “In most cases synergy was only reported at antimicrobial concentrations above established breakpoints of resistance or the concentration at which synergy was observed was not reported”.